# Effects of supplemental oxygen on systemic and cerebral hemodynamics in experimental hypovolemia: Protocol for a randomized, double blinded crossover study

Sole Lindvåg Lie[1,2,3]*, Jonny Hisdal[2,3], Marius Rehn[1,4,5], Lars Øivind Høiseth[1,6]

1 Department of Research and Development, Norwegian Air Ambulance Foundation, Oslo, Norway,
2 Faculty of Medicine, University of Oslo, Oslo, Norway, 3 Section of Vascular Investigations, Oslo University Hospital, Oslo, Norway, 4 Division of Prehospital Services, Air Ambulance Department, Oslo University Hospital, Oslo, Norway, 5 Faculty of Health Sciences, University of Stavanger, Stavanger, Norway, 6 Division of Emergencies and Critical Care, Department of Anesthesiology, Oslo University Hospital, Oslo, Norway

* sole.lindvaag.lie@norskluftambulanse.no

**Data Availability Statement:** No datasets were generated or analysed during the current study. All

## Abstract

Supplemental oxygen is widely administered in trauma patients, often leading to hyperoxia. However, the clinical evidence for providing supplemental oxygen in all trauma patients is scarce, and hyperoxia has been found to increase mortality in some patient populations. Hypovolemia is a common finding in trauma patients, which affects many hemodynamic parameters, but little is known about how supplemental oxygen affects systemic and cerebral hemodynamics during hypovolemia. We therefore plan to conduct an experimental, randomized, double blinded crossover study to investigate the effect of 100% oxygen compared to room air delivered by a face mask with reservoir on systemic and cerebral hemodynamics during simulated hypovolemia in the lower body negative pressure model in 15 healthy volunteers. We will measure cardiac output, stroke volume, blood pressure, middle cerebral artery velocity and tolerance to hypovolemia continuously in all subjects at two visits to investigate whether oxygen affects the cardiovascular response to simulated hypovolemia. The effect of oxygen on the outcome variables will be analyzed with mixed linear regression. The study is registered in the European Union Drug Regulating Authorities Clinical Trials Database (EudraCT, registration number 2021-003238-35).

## Introduction

Supplemental oxygen is frequently administered in acutely and critically ill patients to avoid arterial hypoxemia and tissue hypoxia [1]. For trauma patients, this is stated in the ATLS (Advanced Trauma Life Support) guidelines: "*Supplemental oxygen must be administered to all severely injured trauma patients*" [2]. Accordingly, supplemental oxygen is often given to trauma patients, frequently resulting in hyperoxia [3]. However, the clinical evidence for providing supplemental oxygen in all trauma patients is scarce [4] and the liberal use has been

relevant data from this study will be made available upon study completion.

**Funding:** This study is funded by the Norwegian Air Ambulance Foundation and Oslo University Hospital. The funders had and will not have a role in study design, data collection and analysis, decision to publish, or preparation of the manuscript.

**Competing interests:** The authors have declared that no competing interests exist.

largely founded on a presumption that supplemental oxygen is harmless. There is an increasing focus on possible deleterious effects of hyperoxia [1], and a recent retrospective cohort study on trauma patients receiving supplemental oxygen found higher mortality rates in patients with a higher $SpO_2$ [5].

In the initial treatment of trauma patients, detection and treatment of hypovolemia is of paramount importance. The overriding goal for the resuscitation of these patients is to ensure adequate oxygen delivery to the vital organs, which is given by the product of cardiac output and arterial oxygen content. Hypovolemia leads to reduced cardiac filling, stroke volume and cardiac output [6]. Under normal circumstances in unanesthetized humans, this is compensated by an increase in systemic vascular resistance and heart rate to maintain a normal or near-normal mean arterial pressure (MAP). Normobaric hyperoxia induces vasoconstriction and reduced blood flow to several organs in normovolemic healthy volunteers, including the brain, heart and skeletal muscle [7,8]. Accordingly, hyperoxia may lead to an increased tolerance to hypovolemia mediated by vasoconstriction and thereby maintained MAP as well as a potential increase in arterial oxygen content. However, hyperoxia may lead to reduced tolerance to hypovolemia due to reduced cerebral blood flow.

There is a lack of studies investigating the effect of supplemental oxygen on systemic hemodynamics during hypovolemia in a controlled, experimental setting. We therefore plan to conduct an experimental, randomized, double blinded crossover study where healthy subjects will inhale 100% oxygen or room air administered on a face mask with reservoir during simulated hypovolemia in the lower body negative pressure (LBNP) model. LBNP is an experimental model of central hypovolemia where blood is redistributed from the upper to the lower body [9]. While the separated effects of hyperoxia and LBNP on healthy volunteers are described previously [7,9], the potential effects of hyperoxia on the hemodynamic response to LBNP need elucidation. The aim of the present study is therefore to investigate the effect of supplemental oxygen on systemic and cerebral hemodynamics during LBNP.

The primary hypothesis of this study is that supplemental oxygen will induce a different response in cardiac output compared to room air during LBNP. Secondary hypotheses are that supplemental oxygen induces different responses in stroke volume, middle cerebral artery velocity (MCAV) or time to decompensation during LBNP.

## Materials and methods

### Organization and conduct

The study protocol is written according to the Norwegian Clinical Research Infrastructure Network (NorCRIN) guidelines and registered in the European Union Drug Regulating Authorities Clinical Trials Database (EudraCT, registration number 2021-003238-35). The Norwegian Medical Agency (21/15284-9) and the Regional Ethics Committee (REK South East D, ref. 285164) have assessed and approved the protocol. The original protocol is found in the supporting information file "S1 Protocol" and the spirit checklist in "S1 Checklist".

The sponsor of this trial is Oslo University Hospital, Norway. Experiments will be conducted at The Section of Vascular Investigations, Oslo University Hospital, Oslo, Norway. We will obtain written informed consent from all subjects before the start of the study.

### Design

In this single-center, experimental, randomized, double blinded, crossover trial we will study the effects of supplemental oxygen on systemic and cerebral hemodynamics during simulated hypovolemia in 15 healthy subjects. The schedule of enrolment, interventions, and assessments is shown in **Fig 1**, and study design is illustrated in **Fig 2**. All subjects will participate on two

| | STUDY PERIOD | | | |
|---|---|---|---|---|
| | Enrolment | Post-allocation | | Notes |
| **TIMEPOINT** | $-t_1$ | $t_1$ | $t_2$ | |
| **ENROLMENT:** | | | | |
| **Eligibility screen** | X | | | |
| **Informed consent** | X | | | |
| **Medical history** | X | | | |
| **Physical examination** | X | | | |
| **Highly sensitive urine pregnancy test (WOCBP only)** | X | | | |
| **Vital signs** | X | | | |
| **Allocation/ randomization** | X | | | |
| **INTERVENTIONS:** | | | | |
| **LBNP** | | ←——————→ | | Lower body negative pressure |
| **100% oxygen, 15 L/min** | | X* | | *The order of which 100% oxygen and room air is given depends on the randomization |
| **Room air, 15 L/min** | | | X* | |
| **ASSESSMENTS:** | | | | Complete list of measurements is found in the Outcome Measures section |
| **Cardiac output** | | ←——————→ | | |
| **Stroke volume** | | ←——————→ | | |
| **MCAV** | | ←——————→ | | Middle cerebral artery velocity, cerebral blood flow surrogate. |
| **Time to decompensation** | | ←——————→ | | Decompensation is given by **Table 1.** Stop-criteria |

**Fig 1. SPIRIT schedule of enrolment, interventions, and assessments.**

different visits, with at least one day between each visit. On both visits the subjects will be exposed to LBNP and inhale either 100% oxygen or room air, in a block-randomized order. Except from the inhalation gas, the experiments on Visit 1 and 2 are identical.

Prior to the start of the experiment on either visit, the subject will be familiarized to the set-up and rest for 20–30 minutes in the supine position to stabilize hemodynamic parameters before data sampling begins. After a baseline period, a 5 min run-in time for the inhalation gas will follow. The subjects will be exposed to stepwise LBNP starting at 0 mmHg with 10 mmHg increments every 3 minutes until reaching LBNP 80 mmHg or aborting the experiment (see

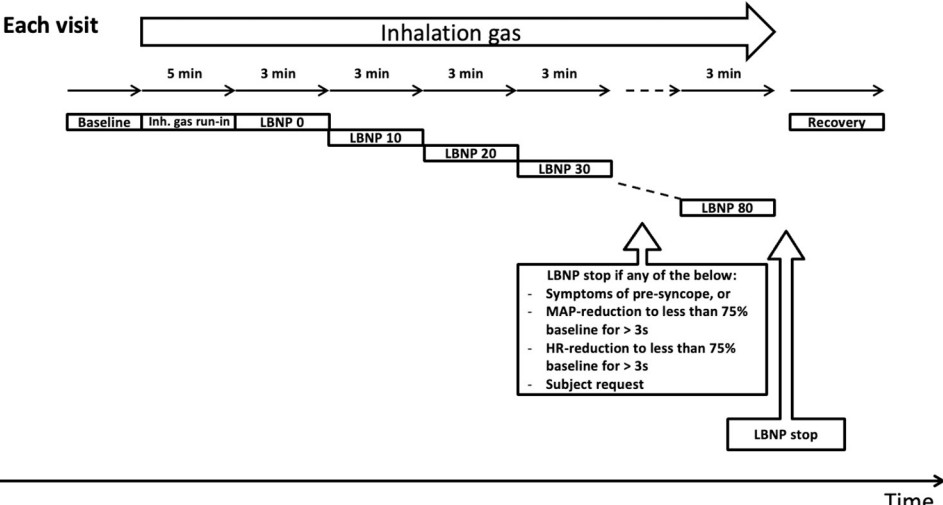

**Fig 2. Schematic illustration of the study design per visit.** The subject receives either 100% oxygen or room air as inhalation gas on Visit 1, and the other on Visit 2, throughout the entire experiment. Lower body negative pressure (LBNP) is increased stepwise with increments of 10 mmHg every 3 min from 0 mmHg until reaching 80 mmHg or aborting. Inh. gas = inhalation gas, MAP = mean arterial pressure, HR = heart rate.

Table 1). As all subjects receive both oxygen and room air, they will act as their own controls due to the crossover design. Both the subjects and the investigators will be blinded to the inhalation gas.

## Randomization

At enrolment, subjects will be randomly assigned (block randomization) in a 1:1 ratio to receive oxygen or room air on Visit 1, and the other on Visit 2. To get at balanced design, the subjects will be randomized with permuted blocks of size 4 or 6, using the "blockrand" package [10] in R [11] /Rstudio [12]. The randomization list will be automatically generated by the principal investigator as a.pdf-document and handed to a 3rd party who will prepare hosing for oxygen or room air administration and envelopes for emergency unblinding. Randomization lists will not be available to the investigators collecting data until after end of the study. Each subject will be dispensed blinded study intervention.

## Eligibility criteria

Subjects will be recruited according to the inclusion and exclusion criteria given in Table 2. In addition, subjects must abstain from caffeine containing products for 6 hours before each visit, nicotine containing products for 12 hours before each visit and strenuous exercise for 3 hours before each visit. Subjects are allowed to have a light meal on the day of the experiment before the experiment begins. Due to potential effects of circadian rhythm on the hemodynamic response to LBNP, we will to the extent possible conduct both visits at a similar time of the day for each subject. However, the evidence supporting the effect of circadian rhythm on the hemodynamic response to LBNP in the literature seems weak [13].

## Interventions

**Lower body negative pressure.** LBNP is a method to simulate central hypovolemia where negative pressure is applied to the body from the waist-down [9] as shown on Fig 3. Thereby,

**Table 1. Stop-criteria.**

| Stop-criteria |
|---|
| Symptoms or signs of impending circulatory collapse |
| • Symptoms of pre-syncope |
| 1. Light-headedness |
| 2. Nausea |
| 3. Sweating |
| • Occurrence of hemodynamic thresholds preceding circulatory collapse (determined from measurements at baseline) |
| 1. MAP-reduction to less than 75% of baseline values (measured at normovolemia) for >3 s |
| 2. HR-reduction to less than 75% baseline values (measured at normovolemia) for >3 s |
| Subject request for reasons other than above |

MAP = mean arterial pressure, HR = heart rate.

blood is displaced from the central compartment of the upper body to the lower extremities and pelvis. The subject is placed in the supine position in the LBNP chamber which is sealed just above the iliac crest. The model has been used for more than half a century and is considered a safe and useful model for studying hypovolemia in conscious volunteers.

**Inhalation gas: Oxygen and room air.** At each visit a subject will inhale either 100% oxygen or room air during the entire experiment as shown in Fig 2. The inhalation gas will be administered on a face mask with reservoir from a gas cylinder connected to a flow meter to ensure an output flow of 15 L/min.

Administration of normobaric oxygen at 100% is not recommended for >6 h due to formation of reactive oxygen species (ROS) [14] and their possible side-effects, primarily affecting the lungs. During the study, administration of 100% oxygen will in most subjects be limited to approximately 30 min, and never exceed 60 min. In essence, we are not aware of significant medical risks with the short-term use of oxygen in healthy adults. There are no absolute contraindications to normobaric oxygen supplementation [14].

**Table 2. Inclusion and exclusion criteria.**

| Inclusion criteria |
|---|
| Age ≥ 18 and < 50 years at the time of signing the consent |
| Overtly healthy as determined by medical evaluation including medical history, heart and lung auscultation, focused cardiac ultrasound and measurement of cardiac conduction times |
| Woman of childbearing potential (WOCBP) must 1) use adequate birth control* or 2) have a negative pregnancy test less than 14 days before visit |
| Capable of giving a signed informed consent |
| **Exclusion criteria** |
| Any medical condition limiting physical exertional capacity or requiring regular medication (allergy and contraceptives excepted) |
| Pregnancy |
| Breastfeeding |
| History of syncope (syncope of presumed vasovagal nature with known precipitating factor excepted) |
| Any known cardiac arrhythmia |
| Any drug (contraceptives excepted) used on a regular basis for a chronic condition (allergy excepted) |

*See "S1 Protocol" for specific requirements to adequate birth control.

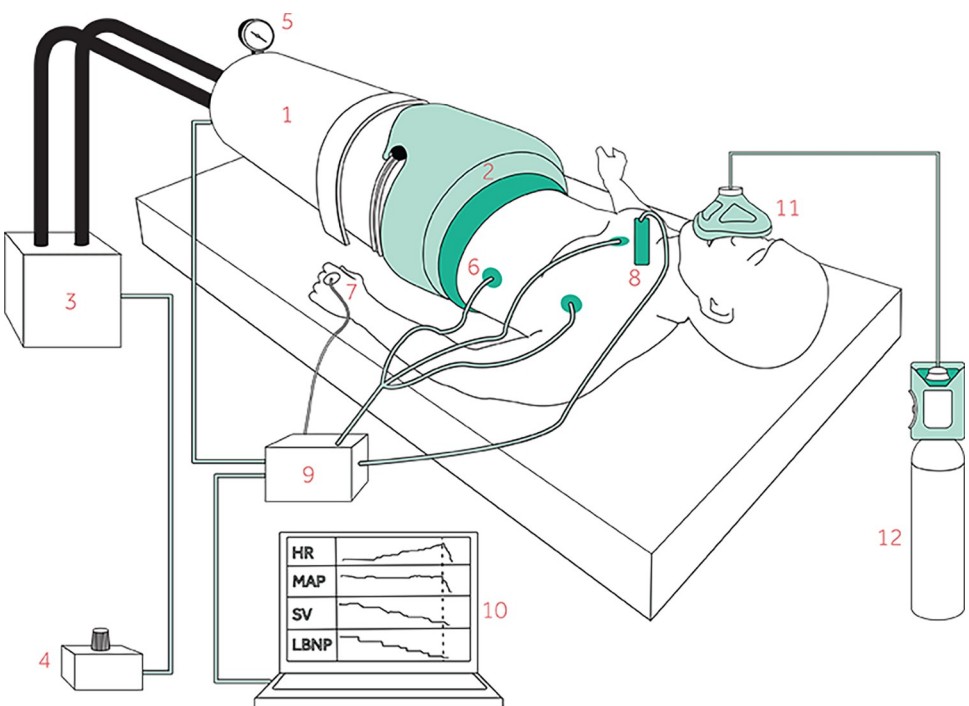

**Fig 3.** Illustration showing the test subject inside 1) the lower body negative pressure (LBNP) chamber. The chamber is 2) sealed just above the iliac crest and connected to 3) a vacuum pump controlled by 4) a pressure control unit. The applied negative pressure is displayed on 5) a pressure monitor. Measurements such as 6) ECG for heart rate (HR), 7) mean arterial pressure (MAP) and 8) stroke volume (SV) are connected to 9) a data acquisition device and 10) sampled on a laptop continuously. The inhalation gas is administered on 11) a face mask connected to 12) a gas cylinder.

## Outcome measures

During each visit we will measure heart rate with a three-lead ECG (Powerlab; ADInstruments, Dunedin, New Zealand). MAP and cardiac stroke volume will be measured with the volume-clamp method on the third finger of the left hand (Nexfin; Edwards Lifesciences corp., CA, USA) and by suprasternal Doppler ultrasound (SD-50 (SD-50; Vingmed Ultrasound, Horten, Norway). Cardiac output is calculated as the product of stroke volume from the Doppler ultrasound and heart rate from the ECG. Middle cerebral artery velocity (MCAV) will be measured using triplex ultrasound (GE E95; General Electric/ Vingmed, Horten, Norway) as a surrogate for cerebral blood flow. Arterial pulse oximetry will be obtained (Masimo Radical 7; Maximo corp., CA, USA) in addition to cerebral oxygen saturation by near infrared spectroscopy (Invos 5100C cerebral/ somatic oximeter; Somanetics, Troy, MI, USA). We will use laser Doppler flowmetry to measure acral skin blood flow (PeriFlux 4001 Master; Perimed AB, Järfälla, Sweden), and volumetric capnography to measure respiratory frequency and end-tidal $CO_2$ (Medlab CAP 10; Medlab GmbH, Stutensee, Germany). Tolerance to hypovolemia will be estimated as time from the start of LBNP 0 to hemodynamic decompensation, where decompensation is defined by **Table 1.** Stop-criteria. All data will be sampled continuously and stored on the hospital's secured server. Our primary and secondary objectives with corresponding endpoints are shown in **Table 3**.

## Discontinuation of study

LBNP is released after 3 min at LBNP 80 mmHg or sooner by occurrence of any of the stop-criteria in given in **Table 1**. For safety reasons, an envelope containing a paper stating the

**Table 3. Primary and secondary objectives and endpoints.**

| Objectives | Endpoints |
| --- | --- |
| **Primary** | |
| Study the effect of supplemental oxygen on cardiac output during LBNP | Difference in the change in cardiac output between oxygen and room air during LBNP |
| **Secondary** | |
| Study the effect of supplemental oxygen on cardiac stroke volume during LBNP | Difference in the change in cardiac stroke volume between oxygen and room air during LBNP |
| Study the effect of supplemental oxygen on MCAV during LBNP | Difference in the change in MCAV between oxygen and room air during LBNP |
| Study the effect of supplemental oxygen on time to hemodynamic decompensation during LBNP | Difference in time to decompensation between oxygen and room air during LBNP |

LBNP = lower body negative pressure, MCAV = middle cerebral artery velocity.

given inhalation gas will we present at all visits for the purpose of emergency unblinding due to medical considerations.

## Sample size

The estimated effect of LBNP on cardiac output with its standard deviation was estimated from the raw data from a previous study [15]. A change of 15% in cardiac output is often used as a threshold when evaluating interventions to increase cardiac output [16]. We assume a mean cardiac output of 4.85 ± 1.08 L/min at baseline, and a change of -0.489 L/min for each LBNP-level (Δ-20 mmHg/level). Error within subjects is assumed independent between LBNP-levels with SD 0.385 L/min. Assuming that a 15% reduction in cardiac output during oxygen inhalation compared to air is significant, this would give an increased reduction (interaction effect) of 0.18 L/min per LBNP level. If assuming a SD of 0.18 L/min for this interaction effect, including 15 subjects would give a $1-\beta = 0.87$ to detect this effect with $\alpha = 0.05$, based on simulations.

## Trial oversight

This study will be monitored by the Clinical Trials Unit (CTU) at Oslo University Hospital to ensure all procedures follow Good clinical practice (GCP) guidelines. Adverse events (AEs) and serious adverse advents (SAEs) will be collected from the start of the experiment on Visit 1 and until the end of Visit 2. All SAEs will be recorded and reported to the sponsor or designee immediately. The investigator will submit any updated SAE data to the sponsor within 24 hours. Fatal or life threatening suspected unexpected adverse reactions (SUSARs) will be reported to The Norwegian Medicines Agency within 7 days, and other SUSARs within 15 days.

## Statistical methods

The effect of oxygen on cardiac output will be analyzed in a mixed linear regression model to account for repeated measurements within subjects. The effect of oxygen on MCAV and cardiac stroke volume will be analyzed in a similar fashion. LBNP-tolerance (time to decompensation) will be analyzed in a mixed proportional hazards model. No interim analysis will be performed.

## Discussion

There are few experimental studies investigating the effect of supplemental oxygen on systemic hemodynamics during simulated hypovolemia. This is unfortunate since trauma patients

often receive supplemental oxygen and may suffer from hypovolemia. To our knowledge, only one study has previously exposed healthy volunteers to LBNP and 100% oxygen while measuring systemic hemodynamics [17]. They found no difference in hemodynamic response to LBNP between 100% oxygen and room air. A limitation to this study was that the authors only applied one level of LBNP, which was also low to moderate (-40 mmHg). In our planned study we will use graded LBNP from 0 to -80 mmHg to induce a greater span of hypovolemia and also estimate cerebral blood flow. We hope that our results can contribute to the understanding of the effect of oxygen on systemic and cerebral hemodynamics during hypovolemia.

When designing this study, we had to weigh the duration of each LBNP-level against the desire to reach a sufficiently high (negative) level of LBNP. By increasing the duration of each LBNP-level, we could potentially increase the number of decompensations at the cost of fewer observations at high LBNP-levels. Based on the decompensation rate in prior work [15,18], we believe that the present LBNP protocol will be able to reveal an effect of oxygen on time to decompensation, i.e. LBNP tolerance. We also believe that the MAP stop-criterion of 25% below baseline values is suitable to detect hemodynamic decompensation, as the change relative to the individual subject's habitual blood pressure is considered. Also, a MAP reduction to less than 75% of baseline values largely coincides with a substantial reduction in systolic blood pressure using an absolute threshold of e.g. 80 mmHg.

There are a few considerations regarding the validity of the suprasternal Doppler ultrasound which is used to measure our main outcome variable. The velocity profile in the ascending aorta is rectangular and preserved for the first 3 cm distal to the aortic orifice, even if the aortic diameter changes [19]. Consequently, slight changes in sample volume location in either lateral or caudal direction will have minor influence on the obtained velocity. In addition, since the suprasternal ultrasound probe is pointed in a craniocaudal direction, a theoretical caudal displacement of the heart with LBNP should have negligible influence on the angle of insonation and hence the obtained velocity.

## Trial status

The study is planned to enroll test subjects from December 2021 to June 2022.

## Supporting information

**S1 Checklist.**
(DOC)

**S1 Protocol.**
(PDF)

## Author Contributions

**Conceptualization:** Sole Lindvåg Lie, Jonny Hisdal, Marius Rehn, Lars Øivind Høiseth.

**Funding acquisition:** Jonny Hisdal, Marius Rehn, Lars Øivind Høiseth.

**Methodology:** Sole Lindvåg Lie, Jonny Hisdal, Marius Rehn, Lars Øivind Høiseth.

**Project administration:** Sole Lindvåg Lie, Lars Øivind Høiseth.

**Supervision:** Jonny Hisdal, Marius Rehn, Lars Øivind Høiseth.

**Writing – original draft:** Sole Lindvåg Lie.

**Writing – review & editing:** Jonny Hisdal, Marius Rehn, Lars Øivind Høiseth.

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
