## [Decision Letter · Decision Letter 0]

22 Feb 2022

PONE-D-21-37763Effects of supplemental oxygen on systemic and cerebral hemodynamics in experimental hypovolemia: Protocol for a randomized, double blinded crossover studyPLOS ONE

Dear Dr. Lindvåg Lie,

Thank you for submitting your manuscript to PLOS ONE. After careful consideration, we feel that it has merit but does not fully meet PLOS ONE’s publication criteria as it currently stands. Therefore, we invite you to submit a revised version of the manuscript that addresses the points raised during the review process.It is not clear how hyperoxia would alter variables influencing cardiac output during LBNP and need a mechanistic rationale to justify the primary outcome variable.Two potential sources of error for the primary variable mentioned by reviewer need to be confirmed and discussed.Need a hypothesis statement.Please submit your revised manuscript by Apr 08 2022 11:59PM. If you will need more time than this to complete your revisions, please reply to this message or contact the journal office at plosone@plos.org. Please include the following items when submitting your revised manuscript:A rebuttal letter that responds to each point raised by the academic editor and reviewer(s). You should upload this letter as a separate file labeled 'Response to Reviewers'.A marked-up copy of your manuscript that highlights changes made to the original version. You should upload this as a separate file labeled 'Revised Manuscript with Track Changes'.An unmarked version of your revised paper without tracked changes. You should upload this as a separate file labeled 'Manuscript'.

We look forward to receiving your revised manuscript.

Kind regards,

Quan Jiang, Ph,D.

Academic Editor

PLOS ONE

Journal Requirements:

Reviewers' comments:

Reviewer's Responses to Questions

**Comments to the Author**

1. Does the manuscript provide a valid rationale for the proposed study, with clearly identified and justified research questions?

Reviewer #1: No

Reviewer #2: Yes

2. Is the protocol technically sound and planned in a manner that will lead to a meaningful outcome and allow testing the stated hypotheses?

Reviewer #1: No

Reviewer #2: Yes

3. Is the methodology feasible and described in sufficient detail to allow the work to be replicable?

Reviewer #1: No

Reviewer #2: Yes

4. Have the authors described where all data underlying the findings will be made available when the study is complete?

Reviewer #1: No

Reviewer #2: Yes

5. Is the manuscript presented in an intelligible fashion and written in standard English?

Reviewer #1: Yes

Reviewer #2: Yes

6. Review Comments to the Author

You may also provide optional suggestions and comments to authors that they might find helpful in planning their study.

Reviewer #1: General concerns with the protocol:

Based upon the text in lines 66-70, I presumed that the primary variable of interest was tolerance to LBNP. However, Table 3 indicates that this variable is a secondary observation, with differences in the change in cardiac output between trials being the primary observation. This is perplexing given the absence of a physiologically-sound rationale for the mechanisms by which hyperoxia would alter cardiac output during LBNP. Cardiac output changes during LBNP are primarily mediated by reductions in venous return, coupled with baroreflex-driven withdrawal of cardiac vagal tone (affecting heart rate) and accompanying increases in cardiac sympathetic activity (affecting heart rate and contractility). Given this, it is not clear how hyperoxia would alter variables influencing cardiac output during LBNP. A mechanistic rationale should be proposed justifying the primary outcome variable.

I struggle with the assumption that the diameter of the aorta for cardiac output calculations does not change throughout LBNP. There are two possible reasons why that diameter may in fact change. 1) It is highly unlikely that the location of the ultrasound beam (ideally the aortic root, which is not verified and thus unknown) would remain stable throughout LBNP. Possible factors that could influence the location of this beam include subtle angle changes of the probe associated with user “variability”, gradual position changes as the participant is pulled into the LBNP chamber, and/or possible changes in the position of the heart during LBNP. 2) Prior work has shown that low levels of LBNP (e.g., 40 mmHg) change the diameter of the ascending aorta (see PMID 7776239). Given that it is unlikely that the ultrasound beam is consistently focused on the aortic root, there is a high likelihood that LBNP itself is reducing the diameter of the assessed area. Both issues are critical given the substantial error that small differences in aortic diameter have on calculations of stroke volume, and thus cardiac output. As an example, a 10% error in the diameter of the aorta (e.g., 18 mm actual diameter rather than a proposed 20 mm fixed diameter) would result in an ~1 l/min error in cardiac output at a heart rate of 60 bpm. These two potential sources of error for the primary variable are very concerning.

Given the author’s prior work showing that approximately 50-75% of participants can tolerate 4.5 min of 80 mmHg LBNP, for the LBNP tolerance question it is unclear why LBNP does not continue until pre-syncope for all participants. If only 25% of the proposed 15 participants (e.g., ~4 participants) achieve pre-syncope at 80 mmHg, it is unlikely that an effect of hyperoxia on LBNP tolerance will be identified.

Specific recommendations:

Introduction:

Please include a hypothesis statement to inform the reader what the authors are proposing will occur during the hyperoxia trial.

Table 1: The MAP and HR “stop-criteria” are of concern for the LBNP tolerance question. In reviewing the authors’ prior work (and associated figures), I didn’t see any evidence of MAP being 75% below pre-LBNP baseline in the individuals who stopped LBNP prior to 80 mmHg. If someone has a blood pressure of 120/80 (mean = 93 mmHg, depending on the formula used), then a blood pressure of 100/55 during LBNP would equal a 75% reduction from pre-LBNP baseline, thus stopping the LBNP trial. In the hundreds of LBNP trials that I’ve conducted, I can’t think of a single occurrence when someone became pre-syncopal at such a high (relatively speaking) blood pressure. Consider using a more accepted blood pressure threshold for pre-syncope such as a sustained systolic blood pressure below 80 mmHg (see PMID: 25071587). I am also perplexed by the HR statement of “to less than 75% baseline values”. Though HR often decreases during LBNP at pre-syncope, HR values less than 75% below baseline almost never occurs (e.g., resting HR of 60 bpm so a HR of 45 bpm would be a stopping criterion).

Line 134-135: Given the impact of meals on gut blood flow, the authors should ensure similar meals are consumed for both trials.

Line 198-200: I believe there is a mistake in this sentence given that LBNP stages will be 10 mmHg every 3 min, as conveyed in figure 2. Also, please provide a citation supporting the statement that cardiac output decreases linearly at a rate of 0.489 L/min for each 20 mmHg LBNP; do we know that this is a linear response (e.g., the same reduction in cardiac output between 0 and 20 mmHg LBNP as between 60 and 80 mmHg LBNP)? Clearly, adding 20 mmHg LBNP from 0 mmHg would be less of a cardiovascular stress relative to adding 20 mmHg LBNP from 60 mmHg LBNP (e.g., from 60 to 80 mmHg).

Line 201: I am confused regarding the statement of a 15% reduction in oxygen in a protocol where 100% oxygen is administered.

Line 219: Wouldn’t LBNP tolerance be assessed between trials via a paired T-test?

Reviewer #2: Manuscript Number: PONE-D-21-37763

Full Title: Effects of supplemental oxygen on systemic and cerebral hemodynamics in experimental hypovolemia: Protocol for a randomized, double blinded crossover study

Thank you for the privilege in reviewing this protocol for the above study. The reviewer works as an academic anaesthetist in a university teaching hospital, with clinical and research interests in an anaesthesia for complex cardiac surgery and liver transplantation. This reviewer has no conflicts of interests.

In summary, this research is unique, commendable, and addresses a clinically important and relevant question. Moreover, this research could provide critical pilot data for the design of larger clinical trial for trauma patients. The safety aspects for the healthy volunteers have been carefully considered and the primary and secondary end points are valid. The study design is thorough. This is an excellent study and the authors are to be congratulated.

Minor points

1. Please comment on the preop fluids allowances and ensure this is built into the protocol and is consistent for all experiments. The authors state that patients may have a lights meal before, but volaemic status is arguably more important to control for, as this could confound the results. Please consider.

2. As a secondary outcome, please consider adding in the measurements of venous blood gases (baseline, mid experiment, end). Translating the changes in biochemical outcomes e.g., base deficit, lactate, hemoglobin, hematocrit, may further strengthen the scientific integrity of this excellent study. A venous blood gas imposes minimal additional risks and there would be little reason to think that these participants would not consent for this small additional step.

3. As an exploratory outcome, please consider whether funding allows for the assessment of markers of endothelial/glycocalyx function (e.g., heparin sulphate, syndecan 1 etc.). This will be an exploratory outcome but may inform the design of a larger clinical trial, especially if a signal is seen between the groups. If the samples are frozen and even analyzed at a later date, this would provide further valuable information about strategies to protect the endothelium during major trauma.

4. Will all experiments be undertaken at the same time of the day? If not, could the hydration status from overnight fasting impact on any of the outcomes, especially if the cross over is such that first experiment for a given participant is conducted in the morning and the second experiment for the same participant in the afternoon. Please comment.

7. PLOS authors have the option to publish the peer review history of their article (what does this mean?). If published, this will include your full peer review and any attached files.

Reviewer #1: No

Reviewer #2: **Yes: **Laurence Weinberg

---

## [Author Response · Author response to Decision Letter 0]

7 Apr 2022

Dear Editor,

Many thanks for the constructive and positive feedback on our manuscript. Please see the revised version addressing all reviewers’ concerns. Below is a list commenting on all reviewers’ concerns and describing the actions we have taken to improve the manuscript. Changes in the manuscript are highlighted with red color. 

Reviewer #1:

Based upon the text in lines 66-70, I presumed that the primary variable of interest was tolerance to LBNP. However, Table 3 indicates that this variable is a secondary observation, with differences in the change in cardiac output between trials being the primary observation. This is perplexing given the absence of a physiologically-sound rationale for the mechanisms by which hyperoxia would alter cardiac output during LBNP. Cardiac output changes during LBNP are primarily mediated by reductions in venous return, coupled with baroreflex-driven withdrawal of cardiac vagal tone (affecting heart rate) and accompanying increases in cardiac sympathetic activity (affecting heart rate and contractility). Given this, it is not clear how hyperoxia would alter variables influencing cardiac output during LBNP. A mechanistic rationale should be proposed justifying the primary outcome variable.

We completely agree with the reviewer regarding the general mechanisms behind changes in cardiac output during LBNP alone. While we also agree that any effect of hyperoxia on the tolerance to hypovolemia is of interest, we believe that the general hemodynamic effects of hyperoxia during LBNP still remain to be sufficiently elucidated, as stated in lines 71-72, before tolerance can be studied more closely. Only one study has investigated the effect of hyperoxia on cardiac output during LBNP and they only applied one level of 40 mmHg of LBNP (https://doi.org/10.1177/147323000803600203). For this reason, effects on changes in cardiac output is reflected in the primary hypothesis of the present study as we believe changes in cardiac output is the better parameter to describe the degree of hypovolemia. We agree with the reviewer that it is not clear how hyperoxia would alter variables influencing cardiac output during LBNP, which is why we will perform the present study. The mechanistic rationale is given by the finding that cardiac output is reduced with hyperoxia during normovolemia in healthy volunteers (lines 64-66). Further, hyperoxia leads to an increase in systemic vascular resistance and accompanying reductions in heart rate and cardiac output, although the mechanisms are not fully understood (https://doi.org/10.1152/ajpregu.00165.2017). We have added a sentence (line 77-79) to clarify the rationale behind the study interventions. Therefore, we think it is of interest to investigate whether hyperoxia changes the cardiovascular response to LBNP with cardiac output as the primary outcome variable.

I struggle with the assumption that the diameter of the aorta for cardiac output calculations does not change throughout LBNP. There are two possible reasons why that diameter may in fact change. 1) It is highly unlikely that the location of the ultrasound beam (ideally the aortic root, which is not verified and thus unknown) would remain stable throughout LBNP. Possible factors that could influence the location of this beam include subtle angle changes of the probe associated with user “variability”, gradual position changes as the participant is pulled into the LBNP chamber, and/or possible changes in the position of the heart during LBNP. 2) Prior work has shown that low levels of LBNP (e.g., 40 mmHg) change the diameter of the ascending aorta (see PMID 7776239). Given that it is unlikely that the ultrasound beam is consistently focused on the aortic root, there is a high likelihood that LBNP itself is reducing the diameter of the assessed area. Both issues are critical given the substantial error that small differences in aortic diameter have on calculations of stroke volume, and thus cardiac output. As an example, a 10% error in the diameter of the aorta (e.g., 18 mm actual diameter rather than a proposed 20 mm fixed diameter) would result in an ~1 l/min error in cardiac output at a heart rate of 60 bpm. These two potential sources of error for the primary variable are very concerning.

We agree with the reviewer and are aware of the possible theoretical limitations of suprasternal Doppler ultrasound to measure cardiac stroke volume. However, for the use of suprasternal Doppler, we refer to PMID 2287179. The velocity profile in the ascending aorta has been shown to be preserved for the first 3 cm distal to the aortic orifice, even if the aortic diameter changes. Further, the velocity profile in the LVOT is rectangular, and not parabolic. Therefore, minor changes in location of the Doppler sample volume both in depth and distance from the aortic wall, will have minor influence on the Doppler measurements. The angle of insonation may change slightly if the heart is displaced laterally, but caudal displacement during LBNP will probably not have a large influence on this angle of insonation as the suprasternal acoustic window gives a very craniocaudal direction of the ultrasound beam. We have addressed this on line 255-262 in the revised manuscript.

In addition to the maintenance of velocity despite aortic diameter changes (as stated above), the reference provided by the reviewer on aortic diameter during LBNP refers to a reduction in pulse area, and not area per se. In this reference, the maximal aortic diameter decreases, but the minimal diameter (at start of systole) increases with LBNP. The diastolic area, which is the area at the beginning of systolic anterior flow thus increases with LBNP, and these effects will therefore to some extent cancel each other out.

Even if changes in aortic diameter and angle would cause a bias in the suprasternal Doppler measurements, we believe one can assume that this effect is comparable between the two visits. As each subject will act as its own control, there is little reason to suspect that these possible sources of error will affect the estimated effect of the intervention between the two visits (within subjects). Also, other non-invasive methods to estimate cardiac stroke volume have their own limitations, with pulse wave analysis (PWA) among others being sensitive to changes in systemic vascular resistance (https://doi.org/10.1016/j.bja.2020.09.049), which does change during LBNP. That being said, we also estimate cardiac stroke volume with PWA in addition to suprasternal Doppler ultrasound in our lab, and plan to present the agreement between these methods in a separate manuscript. 

Given the author’s prior work showing that approximately 50-75% of participants can tolerate 4.5 min of 80 mmHg LBNP, for the LBNP tolerance question it is unclear why LBNP does not continue until pre-syncope for all participants. If only 25% of the proposed 15 participants (e.g., ~4 participants) achieve pre-syncope at 80 mmHg, it is unlikely that an effect of hyperoxia on LBNP tolerance will be identified.

In our latest study using LBNP (https://doi.org/10.1007/s00421-021-04693-6) 11 of 16 (69%) subjects tolerated and finished 6 min of 60 mmHg of LBNP, and only 2 of 16 (13%) completed 80 mmHg of LBNP. We therefore believe that the planned LBNP protocol is suitable to detect potential differences between visits due to hyperoxia. When designing the study, we had to balance the time at each LBNP-level with the desire to reach a sufficiently high (negative) level of LBNP. By increasing the time at each LBNP-level, we could potentially increase the number of decompensations at the cost of fewer observations at high LBNP-levels. We have addressed this on line 245-250 in the revised manuscript. 

Specific recommendations:

Introduction:

Please include a hypothesis statement to inform the reader what the authors are proposing will occur during the hyperoxia trial.

We have added a hypothesis statement at the end of the Introduction section (line 81-84).

Table 1: The MAP and HR “stop-criteria” are of concern for the LBNP tolerance question. In reviewing the authors’ prior work (and associated figures), I didn’t see any evidence of MAP being 75% below pre-LBNP baseline in the individuals who stopped LBNP prior to 80 mmHg. If someone has a blood pressure of 120/80 (mean = 93 mmHg, depending on the formula used), then a blood pressure of 100/55 during LBNP would equal a 75% reduction from pre-LBNP baseline, thus stopping the LBNP trial. In the hundreds of LBNP trials that I’ve conducted, I can’t think of a single occurrence when someone became pre-syncopal at such a high (relatively speaking) blood pressure. Consider using a more accepted blood pressure threshold for pre-syncope such as a sustained systolic blood pressure below 80 mmHg (see PMID: 25071587). I am also perplexed by the HR statement of “to less than 75% baseline values”. Though HR often decreases during LBNP at pre-syncope, HR values less than 75% below baseline almost never occurs (e.g., resting HR of 60 bpm so a HR of 45 bpm would be a stopping criterion).

As previously stated, this study was primarily designed to investigate the hemodynamic response to ongoing hypovolemia, and not hemodynamics at the point of cardiovascular decompensation. Neither was our previous work designed to do so and therefore the figures do not show MAP values at the point of termination of LBNP. In fact, in our previous work, the LBNP-level where decompensation occurred were removed from the analyses (as explained in the manuscripts), as this represents a physiological condition completely different from that of compensated hypovolemia. While the reduction in blood pressure (120/80 to 100/55) suggested by the reviewer represents a 25% reduction in MAP, it also represents an increase in pulse pressure (from 40 to 45), which does not typically occur with LBNP. For the systolic pressure of 100 mmHg presented, one would therefore typically see a higher diastolic pressure than 55 and thereby a higher MAP. 

The range of normal systolic blood pressure (SBP) in men and women from the age 18-50 is quite broad with the 5th percentile being 95 mmHg and the 95th percentile being 160 mmHg (https://doi.org/10.1038/jhh.2013.85). A reduction to <80 mmHg as suggested by the reviewer, may thus represent a reduction of 15 mmHg for some and 80 mmHg for others. Due to the large intraindividual variation in absolute blood pressure, we therefore chose a stop criterion based on a relative reduction, with a 25% reduction in MAP being a substantial reduction. 

Lastly, we find it hard to ethically defend a greater reduction in MAP of more than 25% without aborting the trial. This is also why we included the stop-criteria of 25% reduction in HR, as this consistently indicates decompensation. In essence, our experience is that a reduction of SBP to <80 mmHg in most cases coincides with a reduction of MAP of >25%, but based on the above, we believe a reduction of 25% is appropriate. See line 250-254 in the revised manuscript. 

Line 134-135: Given the impact of meals on gut blood flow, the authors should ensure similar meals are consumed for both trials.

We agree and believe “light meal” as stated in the manuscript is sufficient to provide similar amount of food intake for both visits. 

Line 198-200: I believe there is a mistake in this sentence given that LBNP stages will be 10 mmHg every 3 min, as conveyed in figure 2. Also, please provide a citation supporting the statement that cardiac output decreases linearly at a rate of 0.489 L/min for each 20 mmHg LBNP; do we know that this is a linear response (e.g., the same reduction in cardiac output between 0 and 20 mmHg LBNP as between 60 and 80 mmHg LBNP)? Clearly, adding 20 mmHg LBNP from 0 mmHg would be less of a cardiovascular stress relative to adding 20 mmHg LBNP from 60 mmHg LBNP (e.g., from 60 to 80 mmHg).

As with all statistical models, power and sample size calculations always assume some simplifications. The assumption of the stated decrease in cardiac output is, as stated in the original manuscript, from the raw data for https://doi.org/10.1371/journal.pone.0219154, designated as reference 15. We have clarified this on line 205 in the revised manuscript. The assumption of a linear relationship between LBNP-level and cardiac output seems reasonable given the estimates presented in the figure below (panel bottom right). Although the cardiac stress may be less with a change from 0 to 20 mmHg vs. a change from 60 to 80 mmHg, the relative reduction in cardiac output seems similar, allowing the assumption of a linear relationship. From 0 to 20 mmHg, the reduction in stroke volume seems less than at higher LBNP, but so is the increase in heart rate. After multiplication, this results in a similar change in cardiac output. 

Line 201: I am confused regarding the statement of a 15% reduction in oxygen in a protocol where 100% oxygen is administered.

We apologize for the spelling mistake, which has been corrected.

Line 219: Wouldn’t LBNP tolerance be assessed between trials via a paired T-test?

Time to decompensation between groups could be compared in a paired t-test, but we plan to compare these in a mixed proportional hazards model (mixed Cox regression). This has been specified in the revised manuscript on line 229-230. As previously stated by the reviewer, we will not expect all subjects to decompensate, and these observations will thus be censored at the end of the protocol. To apply a paired t-test, these observations would have to be designated as missing or be given a fixed maximal value. A Cox-regression would however handle these observations as censored. In essence, we believe survival analyses (time-to event) such as Cox regression is well suited to describe time to decompensation. 

Reviewer #2: 

Minor points

1. Please comment on the preop fluids allowances and ensure this is built into the protocol and is consistent for all experiments. The authors state that patients may have a lights meal before, but volaemic status is arguably more important to control for, as this could confound the results. Please consider.

We agree that baseline volaemic status is important and will generally consider healthy volunteers as euvolemic after a light meal. We will also strive to conduct both visits within a subject on the same time of day. 

2. As a secondary outcome, please consider adding in the measurements of venous blood gases (baseline, mid experiment, end). Translating the changes in biochemical outcomes e.g., base deficit, lactate, hemoglobin, hematocrit, may further strengthen the scientific integrity of this excellent study. A venous blood gas imposes minimal additional risks and there would be little reason to think that these participants would not consent for this small additional step.

We agree that biochemical analyses, both “traditional” tests such as those from venous blood gas, or more specific analyses for glycocalyx degradation (below) or possibly tests of oxidative stress, would be of interest. In this study, however, our resources are unfortunately limited to focus on the hemodynamic response. We will keep the suggestions in mind for future studies. 

3. As an exploratory outcome, please consider whether funding allows for the assessment of markers of endothelial/glycocalyx function (e.g., heparin sulphate, syndecan 1 etc.). This will be an exploratory outcome but may inform the design of a larger clinical trial, especially if a signal is seen between the groups. If the samples are frozen and even analyzed at a later date, this would provide further valuable information about strategies to protect the endothelium during major trauma.

Please see our response above. Depending on the results of the present study, we hope to address these questions in future studies. 

4. Will all experiments be undertaken at the same time of the day? If not, could the hydration status from overnight fasting impact on any of the outcomes, especially if the cross over is such that first experiment for a given participant is conducted in the morning and the second experiment for the same participant in the afternoon. Please comment.

As stated above, we will strive to conduct the experiments at the same time of day for each subject. The subjects are told to have “a light meal” before both visits and we will therefore consider them as euvolemic. To account for circadian variations, we will to the extent possible attempt to perform the visits at a similar time of the day for each subject as specified on line 141-144. However, the evidence supporting the effect of circadian rhythm on the hemodynamic response to LBNP in the literature seems weak. To our knowledge, only one study (https://doi.org/10.1152/jappl.1994.76.6.2602) have investigated how time of day affect the response to LBNP. Gillen et al state that preliminary experiments showed a smaller decrease in stroke volume when LBNP was applied in the morning compared with the afternoon, but they do not present data supporting the statement. Also, they show in a subgroup of three subjects that LBNP 40 mmHg in the morning induces a similar decrease in stroke volume as LBNP 30 mmHg in the afternoon (morning ΔSV = 52±3 mL, afternoon ΔSV = 60±6 mL).

---

## [Decision Letter · Decision Letter 1]

14 Jun 2022

Effects of supplemental oxygen on systemic and cerebral hemodynamics in experimental hypovolemia: Protocol for a randomized, double blinded crossover study

PONE-D-21-37763R1

Dear Dr. Lindvåg Lie,

We’re pleased to inform you that your manuscript has been judged scientifically suitable for publication and will be formally accepted for publication once it meets all outstanding technical requirements.

Kind regards,

Quan Jiang, Ph,D.

Academic Editor

PLOS ONE

Additional Editor Comments (optional):

Reviewers' comments:

Reviewer's Responses to Questions

**Comments to the Author**

1. Does the manuscript provide a valid rationale for the proposed study, with clearly identified and justified research questions?

Reviewer #1: Partly

Reviewer #2: Yes

Reviewer #3: Yes

2. Is the protocol technically sound and planned in a manner that will lead to a meaningful outcome and allow testing the stated hypotheses?

Reviewer #1: Partly

Reviewer #2: Yes

Reviewer #3: Yes

3. Is the methodology feasible and described in sufficient detail to allow the work to be replicable?

Reviewer #1: Yes

Reviewer #2: Yes

Reviewer #3: Yes

4. Have the authors described where all data underlying the findings will be made available when the study is complete?

Reviewer #1: No

Reviewer #2: Yes

Reviewer #3: Yes

5. Is the manuscript presented in an intelligible fashion and written in standard English?

Reviewer #1: Yes

Reviewer #2: Yes

Reviewer #3: Yes

6. Review Comments to the Author

You may also provide optional suggestions and comments to authors that they might find helpful in planning their study.

Reviewer #1: Despite the authors' replies, several methodological concerns remain that I believe will adversely affect both the publishability and associated conclusions of this work. That said, I recognize that this is not the ideal forum to debate those concerns.

Reviewer #2: I am satisfied that the detailed responses to the Reviewers have been adequately addressed. The responses to both reviewers have been suitably responded to.

Reviewer #3: I have no statistical comments: randomization, statistical analysis plan, sample size are all clearly delineated.

7. PLOS authors have the option to publish the peer review history of their article (what does this mean?). If published, this will include your full peer review and any attached files.

Reviewer #1: No

Reviewer #2: No

Reviewer #3: No

---

## [Editor Report · Acceptance letter]

16 Jun 2022

PONE-D-21-37763R1 

Effects of supplemental oxygen on systemic and cerebral hemodynamics in experimental hypovolemia: Protocol for a randomized, double blinded crossover study 

Dear Dr. Lindvåg Lie:

I'm pleased to inform you that your manuscript has been deemed suitable for publication in PLOS ONE. Congratulations! Your manuscript is now with our production department. 

Kind regards, 

on behalf of

Dr. Quan Jiang 

Academic Editor

PLOS ONE